# Effects of Low-Frequency Randomly Polarized Electromagnetic Radiation, as Revealed upon Swelling of Polymer Membrane in Water with Different Isotopic Compositions

**DOI:** 10.3390/ma16134622

**Published:** 2023-06-27

**Authors:** Sergey V. Gudkov, Maxim E. Astashev, Ilya V. Baymler, Polina N. Bolotskova, Valery A. Kozlov, Alexander V. Simakin, Minh T. Khuong, Polina A. Fomina, Nikolai F. Bunkin

**Affiliations:** 1Prokhorov General Physics Institute of the Russian Academy of Sciences, 38 Vavilova St., 119991 Moscow, Russia; s_makariy@rambler.ru (S.V.G.); astashev@yandex.ru (M.E.A.); ilyabaymler@yandex.ru (I.V.B.); simakin@kapella.gpi.ru (A.V.S.); 2Department of Fundamental Sciences, Bauman Moscow State Technical University, 2-nd Baumanskaya Str. 5, 105005 Moscow, Russia; bolotskova@inbox.ru (P.N.B.); v.kozlov@hotmail.com (V.A.K.); khuongthuminh@gmail.com (M.T.K.); polja.fomina@gmail.com (P.A.F.)

**Keywords:** polymer membrane, photoluminescence spectroscopy, low-frequency electromagnetic radiation, isotropic submicron-sized particles, anisotropic submicron-sized particles, luminescence quenching

## Abstract

Photoluminescence from the surface of Nafion polymer membrane upon swelling in water under irradiation by electromagnetic waves at a frequency of 100 MHz was studied. In these experiments, natural deionized (DI) water with a deuterium content of 157 ppm and deuterium-depleted water (DDW, deuterium content is 1 ppm) were explored. We have studied for the first time the effect of linearly and randomly polarized low-frequency electromagnetic radiation on the luminescence excitation. To obtain low-frequency electromagnetic radiation with random polarizations, anisotropic solid submicron-sized particles, which result in depolarization effects upon scattering of the initially linearly polarized radiation, were used. We compared two types of colloidal particles: spherically symmetric (isotropic) and elongated (anisotropic). If the radiation is linearly polarized, the intensity of luminescence from the Nafion surface decreases exponentially as the polymer is soaked, and such a behavior is observed both in natural DI water and DDW. When spherically symmetric submicron-sized particles are added to a liquid sample, the luminescence intensity also decreases exponentially upon swelling in both natural DI water and DDW. At the same time, when anisotropic submicron-sized particles are added to DI water, random jumps in the luminescence intensity appear during swelling. At the same time, the exponential decrease in the luminescence intensity is retained upon swelling in DDW. A qualitative theoretical model for the occurrence of random jumps in the luminescence intensity is presented.

## 1. Introduction

The perfluorinated polymer membrane of Nafion is widely used in fuel cells for hydrogen energy. The reason for the use of such membranes in the hydrogen energetics is very high proton conductivity [1]. Nafion’s unique properties are due to a Teflon backbone and perfluorovinyl ether groups terminated with sulfonate groups. The Teflon base of Nafion is highly hydrophobic, while the sulfonate groups are hydrophilic moieties. Spiral channels of 2–3 nm in diameter, filled with water, are formed inside the membrane bulk upon swelling. These channels are negatively charged due to the dissociation of terminal sulfonate groups in accordance with the following reaction (see Ref. [1]):
R—SO_3_H + H_2_O ⇔ R—SO_3_^−^ + H_3_O^+^,(1)


Therefore, protons are attracted towards these channels, but hydroxide anions are repelled from them. In the process of electrolysis of water, the Nafion membrane separates H^+^ and OH^−^ ions in the bulk of water sample, i.e., the recombination of these ions is essentially suppressed, which is the reason for using these membranes in the low-temperature hydrogen elements [2,3,4].

Most of the research has focused on the bulk properties of Nafion, revealed upon swelling in water. The processes, occurring in the bulk liquid close to Nafion surface, were also comprehensively explored, see the monograph [5] and numerous articles cited there. As shown in [5], solid colloidal microspheres are pushed out from the area adjacent to the membrane surface in the water bulk. This area has been termed the exclusion zone (EZ). The exclusion zone size amounts to hundreds of microns, and the property of expelling colloidal micrometer-sized particles from the membrane surface is maintained for several days. As was hypothesized in [5], water inside the EZ has a specific quasi-crystalline structure; this structure was called the fourth phase of water, see [5] for more detail. A recently published review [6] considers a number of alternative models for the physical nature of the EZ. According to this review, the most realistic model of the EZ formation is diffusiophoresis (see Refs. [7,8,9]). As was shown in our more recent review [10], the EZ formation has a different nature. It was claimed that the results of experiments with photoluminescence, Fourier transform IR spectroscopy and dynamic light scattering allow us to claim that the effect of the EZ formation is controlled by the content of deuterium in water, in which Nafion swells. When Nafion swells in natural deionized water (the deuterium content in such water is 157 ± 1 ppm, see Ref. [11]), the polymer fibers unwind from the membrane surface into the bulk of the surrounding liquid, but these fibers are not completely detached from the surface. The size of area, occupied with the unwound fibers in natural deionized water, amounts to 300 μm, which is approximately the EZ length. It also turned out that the effect of unwinding of polymer fibers is absent in deuterium-depleted water (DDW; the deuterium content is 3 ppm).

As shown in review [10], the most effective experimental technique for investigating the effect of unwinding the polymer fibers is photoluminescence spectroscopy in the scheme of the grazing incidence of an optical pump wave on the Nafion surface. This conclusion was made based on the fact that Nafion’s luminescence centers are terminal sulfonate groups, see Ref. [12]. Furthermore, as was shown in [13], when Nafion is immersed in water, the bundles of polymer fibers tend to turn their ends towards the bulk of water. Since the sulfonate groups (the luminescence centers) are localized at the ends of these fibers, i.e., at the Nafion-water interface, the excitation of photoluminescence in the grazing incidence geometry looks quite reasonable. We can expect that luminescence is excited in different manner for unwound and non-unwound polymer fibers. Since the unwound polymer fibers can be considered as soft matter (i.e., these fibers respond to external slight influences very effectively), the question arises of whether it is possible to control the luminescence from the unwound fibers with the help of any external force?

Indeed, in accordance with reaction (1), some part of the polymer fibers in the bulk of water acquires a negative electric charge, localized on the sulfonate groups (luminescence centers). In our previous experiments [10,12], we could not estimate the ratio of negatively charged/neutral sulfonate groups upon immersing the membrane in water. In our experiments with photoluminescence spectroscopy, we studied the luminescence from the surface of dry Nafion (the sulfonate groups are electrically neutral), and from the membrane surface immediately after immersing in water (in this case, some sulfonate groups become negatively charged). It turned out that after immersion in water, the luminescence intensity practically does not decrease, that is, apparently, both neutral and negatively charged sulfonate groups are the luminescence centers. If the transfer of an electron from charged sulfonate group (the luminescence center) to neutral sulfonate group (another luminescence center) occurs during excitation of luminescence, then in accordance with the general principles of excitation and quenching of luminescence (see monograph [14] and articles [15,16,17]), the luminescent state of both centers can be destroyed, which means that luminescence quenches.

As was obtained in our recent work [18], if photoluminescence is excited from the surface of polymer membrane together with simultaneous irradiation of the membrane with one or two counter-propagating ultrasonic waves, directed across the unwound fibers, the random jumps in the luminescence intensity arise in the case of swelling in natural deionized (DI) water. However, if Nafion swells in DDW, there are no random jumps in the luminescence intensity. There are also no jumps in the luminescence intensity in the case of swelling in DI water at irradiating by a single ultrasonic wave. Thus, we can conclude that the random jumps in the luminescence intensity occur providing that the distance between these fibers should somehow change due to some external action, for example, by using two counter-propagating ultrasonic waves.

The question arises of whether it is possible to change the distance between polymer fibers using external electromagnetic radiation? Indeed, the unwound fibers are either electric monopoles (due to dissociation of the terminal sulfonate group, accompanied by detachment of a proton) or electrically neutral. We can assume that a transverse electromagnetic wave with random directions of the electric field vector **E**, vibrating in the membrane plane (this vector is perpendicular to the unwound fibers), should randomly change the distance between the unwound fibers, which is quite similar to the effect of two counter-propagating ultrasonic waves. The present work is devoted to verify this hypothesis. In the experiment described below, linearly polarized and non-polarized low-frequency electromagnetic waves incident on a polymer membrane surface were used. Random polarizations of initially linearly polarized electromagnetic wave were achieved at the expense of scattering the incident radiation by anisotropic solid submicron-sized particles.

## 2. Materials and Methods

### 2.1. Materials

We examined Nafion N117 plates (Sigma Aldrich, St. Louis, MO, USA) of 175 µm thickness with area 1 × 1 cm^2^. Nafion plates were soaked in Milli-Q water with a resistivity of 4 MΩ·cm (DI natural water); measurements were made 1 h after preparation of the liquid samples with the help of Millipore Milli-Q lab water system. The deuterium content of Milli-Q water was 157 ± 1 ppm. Nafion plates were also soaked in deuterium-depleted water (DDW; deuterium content is about 1 ppm), purchased from Sigma Aldrich, St. Louis, MO, USA.

### 2.2. Instrumentation

#### Photoluminescence Study

In this subsection, we briefly describe the experimental protocol. To study the influence of electromagnetic field on the polymer fibers, unwound into the liquid bulk, we first should bear in mind that this radiation should not be absorbed in water. Obviously, light wave in the visible spectral range is not absorbed in water. It is clear, however, that the wave, having a frequency in the visible range, cannot move the unwound polymer fibers due to rather high inertia of these fibers. This is why we decided on electromagnetic radiation at extremely low frequency. Namely, we used a radio signal at frequency *f* = 100 MHz, i.e., at the wavelength *λ* = 3 m. Let us demonstrate that the electromagnetic waves at this frequency are not absorbed in water. Indeed, as is known [19] (see also the recent experimental study [20], devoted to dielectric properties of water at low frequencies), the interaction of electromagnetic wave with a medium is controlled by the complex dielectric permittivity *ε* = *ε*′ − *iε*″, where the real part *ε*′ describes the “ability” of the medium to be polarized by an external electromagnetic field, and the imaginary part *ε*″ stands for the energy loss, associated with absorption of electromagnetic wave and conversion of its energy into heat. For the coefficients *ε*′ and *ε*″ we obtain, see Ref. [19]:(2)$$\varepsilon^{\prime }\left(\omega \right)=\frac{\varepsilon_{S}-\varepsilon_{\infty }}{1+\left(\omega^{2}\tau^{2}\right)}+\varepsilon_{\infty },$$
(3)$$\varepsilon^{\prime \prime }\left(\omega \right)=\frac{\left(\varepsilon_{S}-\varepsilon_{\infty })(\omega \tau \right)}{1+\left(\omega^{2}\tau^{2}\right)}.$$
where *ω* = 2π*f*, *τ* is the rotational diffusion time of water molecules (see [21]), $\tau =\frac{4\pi \eta r^{3}}{kT},$ *ε_S_* is the static permittivity (for water *ε_S_* = 81), *ε*_∞_ is the permittivity in the optical range (for water *ε*_∞_ = 1.77, the square of refractive index), *r* is the molecular radius (for water *r* = 1.38 Å), and *η* is dynamic viscosity (for water under normal conditions *η* = 8.9 × 10^−4^ Pa·s). Thus, we obtain *τ* ≈ 3.3 ps, *ωτ* ≈ 3.3 × 10^−4^, i.e., *ε*″ << *ε*′ ≈ *ε_S_*, and the absorption of the electromagnetic wave at this frequency can be ignored.

To obtain radiation at frequency 100 MHz, we used an N5183A MXG microwave analog signal generator (Agilent, Santa Clara, CA, USA) and whip antenna 100 MHz, Shenzhen Superbat Electronics Co. Ltd. of a length of 737 mm (Shenzhen, China). This antenna evidently emitted a cylindrical wave, polarized along the antenna axis. A R&S^®^FSV30 radio spectrometer, Rohde & Schwarz International GmbH (Muenchen, Germany) was used to control the radiation intensity. In Figure 1 we exhibit an oscillogram from the screen of the R&S^®^FSV30 spectrometer. The emission bandwidth at the FWHM level is approximately equal to 1.25 MHz, i.e., we are dealing with a narrowband radio signal. The radiation intensity at frequency *f* = 100 MHz was equal to 50 mW.

A photograph of the experimental setup for electromagnetic wave irradiation is shown in Figure 2. Radiation at frequency 100 MHz from the whip antenna (1) was directed to the photoluminescence spectroscopy setup (2), covered with black cloth. The distance between the setup and the antenna was approximately 1.5 m. The schematic of the photoluminescence spectroscopy setup is shown in Figure 3.

The probing radiation of a continuous wave UV laser diode (1) (optical pump) at wavelength *λ* = 369 nm was injected into a multimode optical fiber (2) with a diameter ∅ = 100 μm and a numerical aperture *NA* = *n*·sinα = 0.3, where *n* = 1 is refractive index of air andαis the angle of radiation divergence at the exit of the optical fiber in air. The fiber was fixed in a hole, localized in the center of the bottom of a cylindrical cell (3) made of Teflon. The direction of the pump beam coincides with the optical axis in the cell. The cell was filled with a liquid sample, and was thermally stabilized at room temperature (T = 23 °C) with an accuracy of ±0.1 °C with the help of a thermostat T. We investigated the swelling of a square Nafion plate (4) of height *h* = 10 mm and thickness *d* = 175 µm. The plate was installed vertically in parallel to the optical axis (the geometry of grazing incidence of the pump beam). The vertical edges of the Nafion plates were rigidly fixed with two clamps. Since the size of these clamps was much smaller than the width of the Nafion plate, the free boundary conditions were realized for the central region of the Nafion plate (precisely this region was irradiated by the pump wave).

At the beginning of the experiment, a dry (water-free) Nafion plate (4) was placed in an empty cell (3); this plate could be shifted across the optical axis with a stepper motor (8). This way we found the maximum of the luminescence signal; the corresponding spatial arrangement of the Nafion plate was considered to be optimal. When liquid was poured into the cell, the hydrophobic Nafion plate was bent across the optical axis. However, such bending only led to an effective displacement of the Nafion–water interface (this displacement was about 1 mm), but did not lead to a change in the angle of incidence of the pump beam. To find the optimal position of the plate with respect to the optical axis after immersing the Nafion plate in a liquid sample, additional adjustment was carried out by using stepper motor (8). The luminescence radiation was reflected by the inner surface of the cell (Nafion is transparent in the visible range) and collected along the optical axis. This resulted in a significant increase in the measured luminescence signal, which was received by an optical fiber (5), and transmitted to an FSD-8 mini-spectrometer (6) («Optofiber» LLC., Moscow, Russia). Experimental data were accumulated with the help of a computer (7). In this experiment, we studied the temporal dynamics of the luminescence intensity at its spectral maximum (*λ* = 460 nm) as a function of the soaking time *t* of the Nafion plate in the test liquid; we will denote it as *I*(*t*). The start of time counting is related to the moment of pouring the liquid sample into the cell.

The distance between this setup and vertical whip antenna was about 1.5 m. Since the light wavelength at frequency *f* = 100 MHz is *λ* = 3 m, that is, the distance between the setup and whip antenna is about *λ*/2, the direction of the vector **E** along theantenna axis, most likely, will remain unchanged at the distance *λ*/2 from the antenna, i.e., this wave should be linearly polarized in the plane of membrane surface. Since the Nafion plate size (element (4) in Figure 3) was about 1 cm, and the radius of cylindrical wave front at the membrane surface was 1.5 m, we can put that the Nafion plate was irradiated with a linearly polarized wave, having a planar wave front. As will be shown below, it is very important for us that the polarization of the electromagnetic wave along the membrane surface to be random. For this purpose, isotropic (spherically symmetric) and anisotropic (elongated) solid submicron-sized particles were added to the liquid sample. We assumed that these particles should scatter linearly polarized low-frequency incident radiation, resulting (in the case of anisotropic elongated particles) in depolarization of the scattered radiation; the depolarization effects at scattering are described, for example, in the monograph [22].

Solid submicron-sized particles were obtained by using the technique of laser ablation of solid targets in a liquid, see Refs. [23,24,25]. In a recent work [26], a method for obtaining anisotropic submicron-sized particles using iron oxide was described. Spherically symmetric and anisotropic solid submicron-sized particles, obtained with the technique of laser ablation, were investigated by using a Libra 200 FEHR tunneling electron microscope (Carl Zeiss, Oberkochen, Germany). Typical patterns of the submicron-sized particles under study are shown in the photographs Figure 4a–c.

## 3. Experimental Results

In Figure 5, we exhibit the dependence of the luminescence intensity *I*(*t*) at its spectral maximum (*λ* = 460 nm) upon irradiation of the photoluminescence setup with electromagnetic wave at frequency *f* = 99.8 MHz (≈100 MHz), where *t* is the time of soaking the polymer membrane; the moment *t* = 0 is related to immersing the Nafion plate into the liquid sample. Here, we show the graphs of *I*(*t*) in aqueous suspensions containing elongated (anisotropic, Figure 4a) and spherically symmetric gold submicron-sized particles (Figure 4c). These suspensions were prepared on the basis of natural DI water and DDW. In addition, we studied aqueous suspensions of elongated selenium submicron-sized particles (see Figure 4b), prepared on the basis of natural DI water. We also exhibit the dependence of *I*(*t*) for DI water, free of submicron-sized particles.

It is seen that if we are dealing with elongated (anisotropic) solid submicron-sized particles in DI water, random oscillations in the behavior of *I*(*t*) are observed. In this case, there was no reproducibility of the experimental results, that is, there was no point in finding confidence intervals in repeated measurements. If we are dealing with liquid suspension which contained spherically symmetric submicron-sized particles, there are no stochastic oscillations, the luminescence intensity decreases exponentially (see the inset), and the results are fairly well reproduced. In this case, the experimental points correspond to averaging over five measurements; here, we indicate the confidence intervals. If the anisotropic submicron-sized particles were added to DDW, random oscillations were also absent, and the luminescence intensity decreased exponentially, see the inset. Finally, if the submicron-sized particles were not added to the liquid sample (no matter, DI water, or DDW), the stochastic oscillations of *I*(*t*) were not observed as well. It is seen that in the case where the random oscillations are absent, the approximating exponential functions are approximately the same, see the inset.

As will be shown below, the occurrence of random oscillations shows that in some cases, low-frequency electromagnetic radiation can change the distance between the polymer fibers, terminated with the luminescence centers, causing the effects of quenching and, possibly, the enhancement of luminescence.

## 4. Discussion

It is first necessary to explain why we choose the submicron-sized particles of the type, shown in Figure 4. Let us show that for these submicron-sized particles, we can neglect their sedimentation to the bottom of the cell (element (3) in Figure 3) during the entire experiment, i.e., the interaction between the incident low-frequency electromagnetic wave and submicron-sized particles is a stationary process. We estimated by order of magnitude the rate *v* of sedimentation of such particles to the bottom of cell, having a height *H* = 1 cm and filled with water. The estimates were made for selenium submicron-sized particles. We assumed these particles are cylinders with a height *h* = 1 μm and a radius *r* = 10 nm (see Figure 4b). This choice is due to the fact that, according to electron microscopy data, selenium particles are more uniform in size and shape as compared with gold submicron-sized particles, see Figure 4a. The rate *v* of sedimentation can be estimated by using the following formula:(4)$$\frac{4}{3}\pi r_{0}^{3}\left(\rho_{Se}-\rho_{W}\right)g=6\pi \eta r_{0}v.$$
where *ρ_Se_* = 4.8 g/cm^3^ is the density of selenium, *ρ_W_* = 1 g/cm^3^ is the density of water, *η* = 1 mPa·s is the dynamic viscosity of water at room temperature, *g* is the acceleration of gravity, and *r*_0_ is the effective radius of the selenium particle. The effective radius *r*_0_ corresponds to a sphere whose volume is equal to the volume of a cylinder with sizes *r* and *h*. Thus, we obtain $r_{0}=\sqrt[{3}]{\frac{3}{4}r^{2}h}\approx$ 40 nm, and for the sedimentation rate, we have *v* ≈ 1.4 × 10^−9^ m/s. Obviously, for a cell with water, having the height *H* = 1 cm, we can neglect the sedimentation of such particles.

Next, it is necessary to explain why the luminescence intensity, as a rule, decays according to an exponential law, see the magenta, blue, and green curves in Figure 5. The luminescence signal is generally described by an empirical formula, see [12]:(5)$$P=A+kI_{pump}\sigma_{lum}n_{Naf}V.$$
where *I_pump_* is the pump intensity, *A* = 20–270 (in arbitrary units) corresponds to the spectral density of noise of the measuring device, *k* is the apparatus coefficient of the setup, *V* is the luminescent volume, *σ_lum_* is the luminescence cross section, and *n_Naf_* is the volume number density of sulfonate groups (the luminescence centers). Note that Formula (5) does not take into account the lifetime of the luminescent state, since in our experiments, we used continuous wave laser pumping (see Figure 3); in the case of using a pulsed laser, it would be necessary to take into account the lifetime of the luminescent state.

Assuming that the luminescence cross section *σ_lum_= const*, the luminescence intensity *I*(*t*) decreases due to decreasing of the total number of luminescence centers in volume *V*. Since in our case *V* is a constant value (*V* = π*r*^2^*h*, where *r* = 50 μm is the radius of the optical fibers and *h* = 1 cm is the height of the Nafion plate, see Figure 3), the dynamics of *I*(*t*) is controlled by a decrease in the volume number density *n_Naf_* in the near-surface layer of the membrane upon swelling; liquid molecules penetrate into the near-surface layer, resulting in decreasing *n_Naf_*. The dynamics of *n_Naf_*(*t*) obeys the following equation:(6)$$\frac{dn_{Naf}}{dt}=-\frac{n_{Naf}}{\tau },$$
where *τ* is the characteristic time of swelling. Thus, we arrive at:(7)$$n_{Naf}={\left(n_{Naf}\right)}_{0}\exp\left(-\frac{t}{\tau }\right).$$

Since *I*(*t*) ∝ *n_Naf_*(*t*), the luminescence intensity *I*(*t*) at *σ_lum_ = const* decays by an exponential law, as confirmed by the blue, magenta, and green curves in Figure 4. However, if in the process of luminescence excitation, the luminescence center loses an electron, the effect of luminescence quenching may occur, see monograph [14]. Obviously, in this case, the condition *σ_lum_ = const* is violated. Below, we present a scenario of changes in the behavior of *I*(*t*) in the case of *σ_lum_ ≠ const*.

Let us first imagine that the distance *x*_0_ between sulfonate groups, localized at the ends of polymer fibers, unwound in the bulk of liquid, is so small that it becomes possible for an electron to transfer from one sulfonate group (donor) to another one (acceptor), see the diagram in Figure 6. This transfer can occur through a water molecule located between sulfonate groups, which are essentially hydrophilic (see [1]), i.e., water molecules are always present near the sulfonate groups. Note that the electron transfer to a molecule or ion is one of the most common elementary reactions in chemistry, see review [27] for more detail. We should emphasize once again that in our case the donors and acceptors are the luminescence centers localized at the ends of the unwound polymer fibers. To the best of our knowledge, such a model has not previously been theoretically considered.

The transfer of an electron is realized due to crossing of two harmonic (vibrational) potentials. These are the donor potential:(8)$$U_{d}=\frac{m\omega^{2}}{2}x^{2}+U_{0},$$
and the acceptor potential:(9)$$U_{a}=\frac{m\omega^{2}}{2}{\left(x-x_{0}\right)}^{2}+U_{0}.$$

Since the donor and acceptor moieties belong to the same sulfonate groups, let us assume that the initial levels of *U_a_* and *U_d_* are equal to one another: *U_d_*(0) = *U_a_*(*x*_0_) = *U*_0_. Note that in our case of a transferring electron, no new chemical compound is formed; therefore, in Formulas (7) and (8), we do not take into account the thermal effect Δ*H* (the change of enthalpy due to chemical reaction, see [27]). As follows from the model presented in review [27], the wave functions of the orbitals, from which an electron exits and to which it comes, must overlap. Suppose that the functions *U_d_*(*x*) and *U_a_*(*x*) intersect at the point *x’* < *x*_0_: *U_d_*(*x’*) = *U_a_*(*x’*). If the electron energy exceeds the height of the potential barrier (activation energy) *E_A_* = *U_d_*(*x’*) − *U_d_*(0) (see Figure 6), such a transfer becomes possible.

As shown in [27], the source of the activation energy for electron transfer is the orientational polarization of the medium (solvent molecules), as well as the absorption of a light photon. In our case, we can assume that the electron in the potential well *U_d_* acquires energy due to the absorption of the UV wave at the wavelength *λ* = 369 nm, see Section 2. As follows from the graph in Figure 6, the smaller the distance *x*_0_ between the donor and the acceptor, the lower the height of the potential barrier *E_A_*, which must be overcome by an electron to pass from the potential well *U_d_* to the potential well *U_a_*. In accordance with the model presented in review [27], the activation energy of the electron transfer reaction is expressed as:(10)$$E_{A}=\frac{E_{s}}{4},$$
where *E_s_* is the so-called reorganization energy in the process of electron transfer. Assuming that the donor and acceptor are located close to each other, the so-called polarization model of penetrating spheres can be used. In this case, for the energy *E_s_*,we obtain:(11)$$E_{s}=\left(\frac{1}{\varepsilon_{\infty }}-\frac{1}{\varepsilon_{0}}\right)e^{2}\left(\frac{1}{2r_{a}}+\frac{1}{2r_{d}}-\frac{1}{x_{0}}\right),$$
where *ε*_∞_ = 1.77 is the permittivity of water in the optical spectral range, *ε*_0_ = 81 is the static permittivity of water, *r_d_* and *r_a_* are the radii of the donor and acceptor moieties, respectively (in our particular case, obviously, we can assume *r_d_* ≈ *r_a_*), and *x*_0_ is the distance between the donor and acceptor centers, see Figure 6. Note that *x*_0_ is the single controllable parameter here. As follows from (11), the height *E_s_* of the energy barrier drastically decreases as the distance *x*_0_ reduces. In this case, electron transfer from one luminescence center to another one becomes possible, which, in accordance with the data of monograph [14], means that the luminescent state is destroyed and the luminescence is quenched. If the distance *x*_0_ is fixed, the potential barrier height *E_s_* is fixed as well, and the luminescence cross section *σ_lum_*, on average, also does not depend on time. This is realized in the absence of unwinding the polymer fibers into the liquid bulk. At the same time, when polymer fibers are unwound into the liquid bulk, the distance *x*_0_ can vary with time. The same is true for the luminescence cross section: *σ_lum_* appears to be a random function of time. Based on this qualitative model, it is possible to explain the effects associated with irradiation of low-frequency electromagnetic waves.

As shown in our previous work [18], stochastic oscillations in the luminescence intensity *I*(*t*) arise when the polymer membrane is irradiated with counter-propagating ultrasonic waves. These ultrasonic waves were emitted from the opposite sides of the membrane, and were directed towards one another. It is important to note that random oscillations do not occur when unwound polymer fibers are irradiated with a single ultrasonic wave. As was obtained in [18], two ultrasonic waves are absorbed inside a layer of unwound polymer fibers, and in this case, the ultrasonic wave impulse is transferred to the liquid particles, that is, two counter-propagating hydrodynamic flows (acoustic flows) are generated. These flows result in growing the density of the polymer fibers, unwound into the bulk of water, and the maximum density is realized in the central part of the membrane, i.e., in the area that is probed by laser pump; the luminescence is excited precisely in this area. The acoustic flows resulted in a sharp decrease in the distance *x*_0_ between the luminescence centers, decreasing the luminescence intensity *I*(*t*).

As shown in the graphs of Figure 5, in the experiment with a low-frequency electromagnetic wave, random upward and downward jumps in the intensity *I*(*t*) are observed. Note that the decrease in the intensity *I*(*t*) is traditionally explained by the effects of quenching due to the loss of an electron by the luminescence center, see monograph [14]. At the same time, a spontaneous increase in the luminescence intensity, generally speaking, contradicts the law of conservation of energy and occurs either due to an increase in the pump intensity, or due to the addition of special particles—fluorophores, which have a very high luminescence cross section. In our case, both options are excluded. However, within the framework of the suggested theoretical model, both the decrease and the increase in the luminescence intensity can be explained. In fact, the increase in the intensity *I*(*t*) occurs due to the weakening of the quenching effects, that is, due to an increase of *E_s_* in Formula (10). As follows from this formula, the only controllable parameter is the distance *x*_0_ between luminescence centers. In our case, both a spontaneous growth and decrease in the distance *x*_0_ occur. For this, it is necessary that the electric field vector **E** of the low-frequency electromagnetic wave be randomly directed across the unwound fibers, that is, this wave should be randomly polarized.

As noted in Section 2, the radiation from the whip antenna should be linearly polarized, and since the distance between the antenna and the photoluminescence spectroscopy setup is less than the wavelength (see Figure 2), the low-frequency wave on the Nafion surface will save its linear polarization. We do not know whether special devices exist that allow one to change the polarization of a wave at a frequency near 100 MHz. In our experiments, the polarization was changed as a result of scattering by anisotropic submicron-sized particles. In our case, it is necessary to take into account the effects of scattering of a low-frequency wave follows from the graphs in Figure 5. Indeed, there exist clear differences for the green curve (there are no submicron-sized particles) and the black/red curves (anisotropic submicron-sized particles were added to natural DI water). At the same time, there is no influence of the randomly polarized low-frequency electromagnetic wave in DDW, see the blue curve in Figure 5. This can be explained in the frame of our model. Indeed, in the case of DDW there is no unwinding effect of the polymer fibers, and the distance *x*_0_ between fibers is always a fixed value. The fact that the addition of spherically symmetric submicron-sized particles to DI water does not lead to any peculiarities in the dynamics of the intensity *I*(*t*) can be interpreted as the absence of depolarization effects upon scattering by such particles.

In conclusion, we note that the scattering cross section for large wavelengths is very small: σ~λ^−4^, see, for example, [22]. As far as we know, experimental proof of the effects of scattering of meter waves by anisotropic submicron-sized particles was obtained for the first time in this work. Our results obviously require a more rigorous theoretical analysis. However, we can claim that the system, which includes anisotropic scattering particles and polymer fibers, unwound into the water bulk, is very sensitive to external influences of low-frequency electromagnetic waves, which is manifested in experiments with the photoluminescence spectroscopy. It is very important that the observed effects critically depend on the deuterium content in the liquid samples. Despite the fact that the deuterium concentrations of 1 ppm and 157 ppm are very close, the swelling dynamics of the polymer membrane is significantly different for liquids with these deuterium contents. Hence, it seems possible to control the luminescence dynamics when the membrane is irradiated with a randomly polarized electromagnetic wave upon swelling in a liquid with different concentrations of deuterium.

## 5. Conclusions

In this work, we established that the effect of unwinding polymer fibers into the bulk of the surrounding liquid is controlled by the deuterium content in water, and this effect manifests itself at very low deuterium concentrations. Sharp upward and downward jumps in the luminescence intensity occur when unwound fibers are irradiated with a non-polarized low-frequency electromagnetic wave, but are absent when irradiated with a linearly polarized wave. A qualitative theoretical model was proposed; this model is based on the electronic transfer from a donor to an acceptor. In our case, the donors and acceptors are the luminescence centers localized at the ends of the unwound polymer fibers. To the best of our knowledge, such a model has not previously been theoretically considered.

It is known that the loss of an electron by the luminescence center can result in luminescence quenching. However, within the framework of the proposed model, it is possible not only to qualitatively explain arising the sharp downward jumps in the luminescence intensity due to quenching, but also the sharp upward jumps. Assuming that negatively charged/electrically neutral luminescence centers are donors/acceptors of electrons, localized at the ends of unwound fibers, we show that luminescence quenching occurs when these centers approach each other subject to non-polarized electromagnetic wave. At the same time, when the distance between these centers grows, the quenching effect is suppressed, and the luminescence is enhanced.

The random polarization of the incident wave on the surface of a polymer membrane was realized due to the scattering of linearly polarized radiation by anisotropic particles. In fact, in our experiment, we show for the first time that submicron solid particles scatter radiation at a frequency of 100 MHz. Summing up, we found a number of new effects arosedue to unwinding of polymer fibers into the bulk of the surrounding liquid, providing that the membrane is irradiated with a low-frequency non-polarized electromagnetic wave.

## Figures and Tables

**Figure 1 materials-16-04622-f001:**
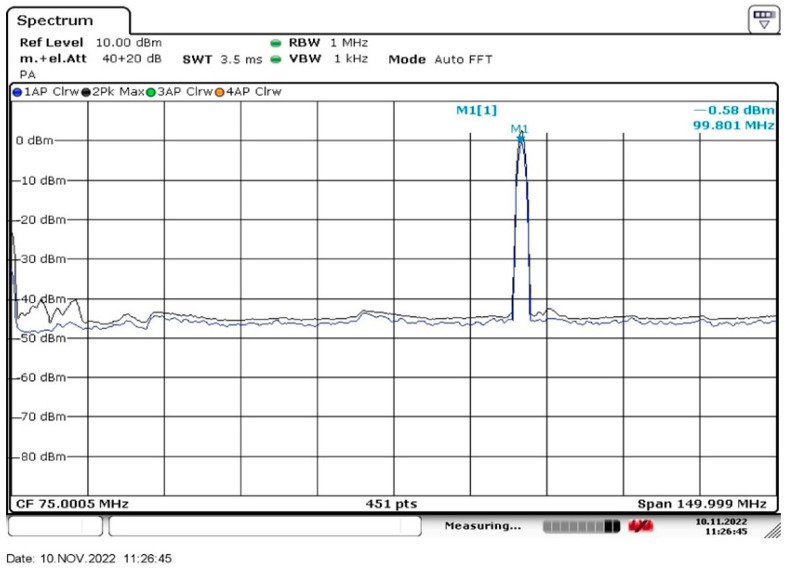
The intensity spectrum of the low-frequency radiation in our experiment.

**Figure 2 materials-16-04622-f002:**
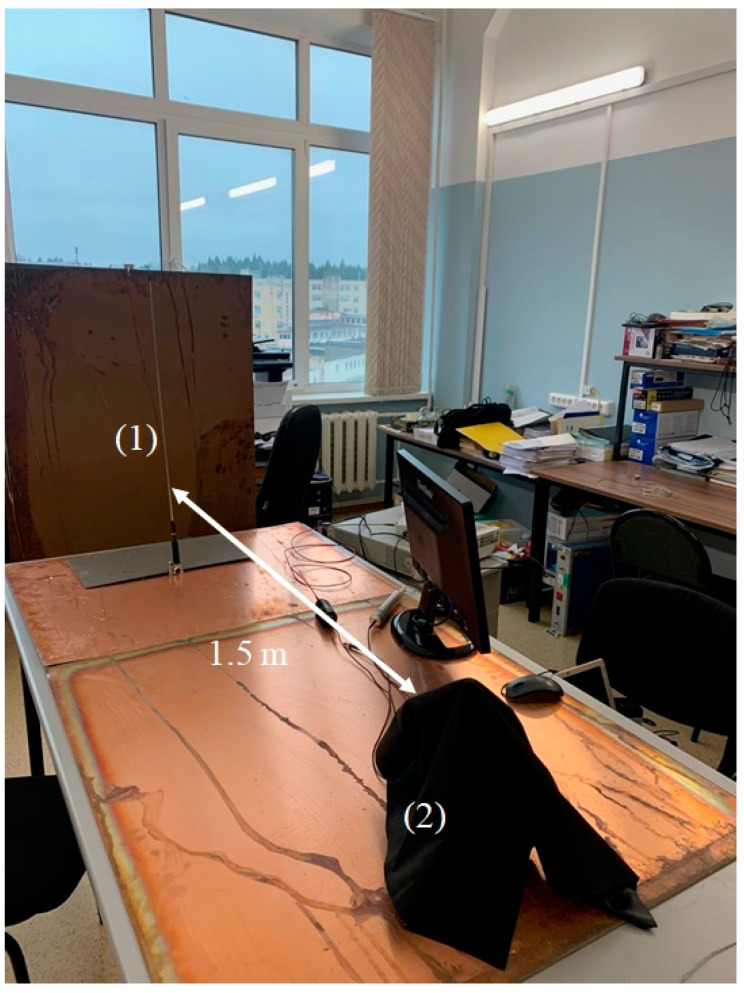
Photograph of the experimental setup. (1)—emitting whip antenna; (2)—setup for photoluminescence spectroscopy, covered with black matter.

**Figure 3 materials-16-04622-f003:**
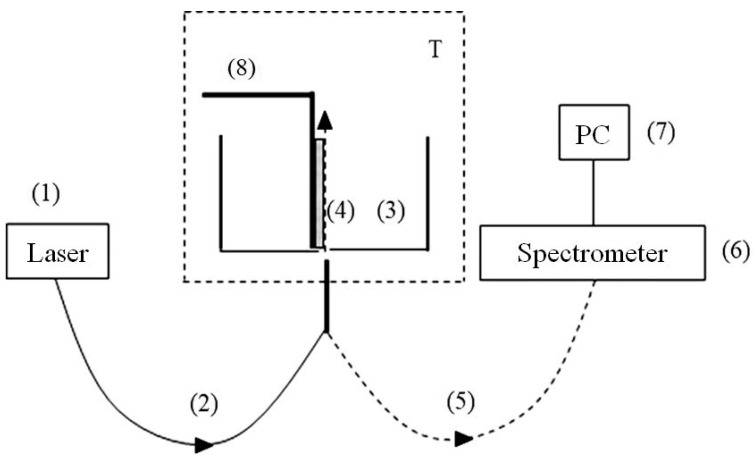
Schematic of the experimental setup for photoluminescence spectroscopy; (1)—laser pump; (2)—multimode quartz optical fiber for transmitting the pump radiation; (3)—cylindrical cell for liquid samples; (4)—Nafion plate; (5)—optical fiber for transmitting the luminescence signal; (6)—mini-spectrometer; (7)—computer; (8)—a stepper motor for the fine adjusting the position of the Nafion plate with respect to the optical axis; T—thermostat.

**Figure 4 materials-16-04622-f004:**
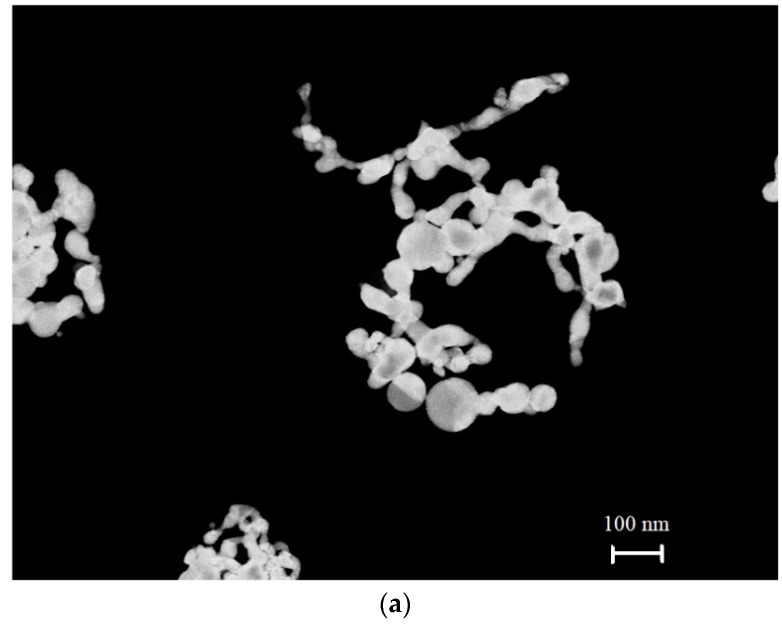
(**a**) Anisotropic (elongated) submicron-sized particles of colloidal gold. (**b**) Anisotropic (elongated) submicron-sized particles of colloidal selenium. (**c**) Spherically symmetric (isotropic) colloidal gold submicron-sized particles.

**Figure 5 materials-16-04622-f005:**
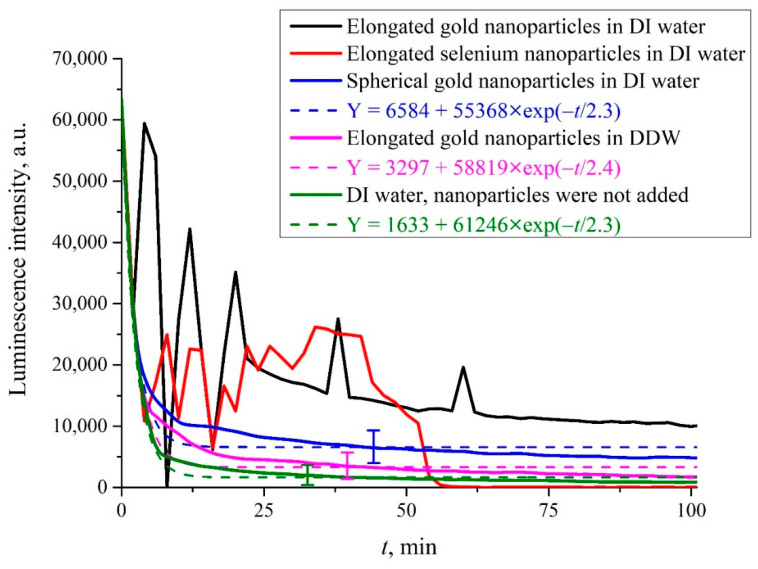
Luminescence intensity *I*(*t*) at the spectral maximum as the function of the soaking time *t* for natural DI water and DDW containing solid submicron-sized particles upon irradiation at frequency *f* = 100 MHz. Black curve: elongated (anisotropic) colloidal gold submicron-sized particles in DI water (Figure 4a). Red curve: elongated (anisotropic) selenium submicron-sized particles in DI water (Figure 4b). Blue curve: isotropic (spherically symmetric) gold submicron-sized particles in DI water (Figure 4c). Purple curve: elongated gold submicron-sized particles in DDW (Figure 4a). Green curve: DI water; submicron-sized particles were not added.

**Figure 6 materials-16-04622-f006:**
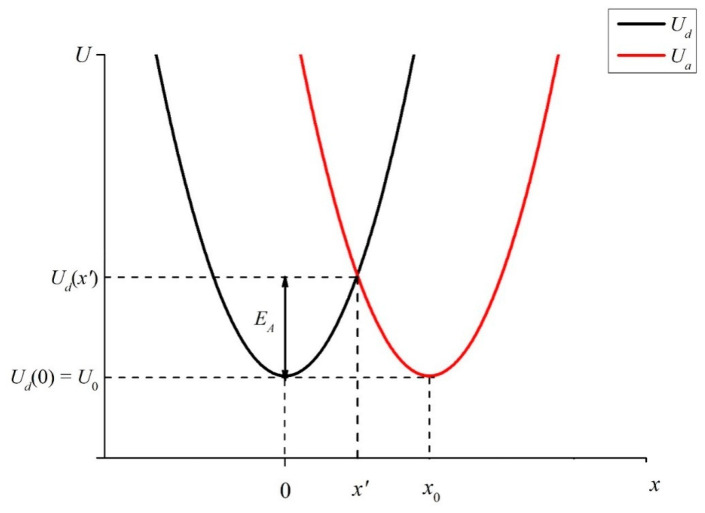
Potentials of vibrational states of electrons for donor *U_d_* and acceptor *U_a_*. The comments are in the text.

## Data Availability

The data presented in this study are available on request from the corresponding author.

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
