# Peer review of "Effects of Low-Frequency Randomly Polarized Electromagnetic Radiation, as Revealed upon Swelling of Polymer Membrane in Water with Different Isotopic Compositions"

_materials, 2023, doi:10.3390/ma16134622_

Round 1

Reviewer 1 Report

The current work focuses on the effects of low-frequency randomly polarized electromagnetic radiation, as revealed upon swelling of polymer membrane in water with different isotopic compositions. The author’s great effort into the manuscript, but minor issues should be addressed. 

-The introduction provides sufficient background and all relevant references are included but the novelty of this work is not highlighted and the author's contribution was unclear compared to other previous works. 

-One of the main problems in the manuscript is that the authors show only results without interpretations of details of it or confirmation by citation. More details are required to explain the obtained results. e.g. Why use only the frequency 100 MHz in the experiment and not use a wide range of frequencies? the used two types of colloidal nanoparticles do not have full identifications and characterization.

Moderate editing of English language required

Author Response

We are grateful to the reviewer for an attentive perusal of our manuscript. The manuscript was completely rewritten in accordance with the reviewer' advices and comments. In what follows we present our replies to particular remarks. Our replies are highlighted with italic font.

-The introduction provides sufficient background and all relevant references are included but the novelty of this work is not highlighted and the author's contribution was unclear compared to other previous works. 

The novelty of the work lies in the fact that we for the first time studied the dynamics of swelling of a polymer membrane, taking into account the unwinding of polymer fibers towards the bulk of the surrounding liquid when the membrane is irradiated with low-frequency radio wave with random polarizations. As far as we know, this problem was formulated by us for the first time. This statement has been emphasized in the new version of the Introduction.

-One of the main problems in the manuscript is that the authors show only results without interpretations of details of it or confirmation by citation. More details are required to explain the obtained results. e.g. Why use only the frequency 100 MHz in the experiment and not use a wide range of frequencies? the used two types of colloidal nanoparticles do not have full identifications and characterization.

We agree with the reviewer that the experimental results require a more detailed theoretical substantiation. However, the theoretical interpretation at the moment seems to be a rather difficult task; new quantum chemical models should be developed for that. At the moment we can only state that the effect of unwinding polymer fibers into the bulk of the surrounding liquid is controlled by the deuterium content in water, and this effect manifests itself in the range of 1–1000 ppm, i.e. at very low concentrations. The studies presented in this paper were a continuation of our earlier experiments, in which we irradiated the unwound polymer fibers with one or two counter-propagating ultrasonic waves. It was found that when irradiated with one ultrasonic wave, there is no effect from such irradiation, while when irradiated with two counter-propagating waves, random jumps in the luminescence intensity occur. It turned out that similar jumps occur when unwound fibers are irradiated with a non-polarized low-frequency electromagnetic wave, but are absent when irradiated with a linearly polarized wave. A qualitative theoretical model was proposed based on the model of chemical reactions due to the transfer of an electron from a donor to an acceptor. In our case, the donors/acceptors are the luminescence centers localized at the ends of the unwound polymer fibers. To the best of our knowledge, such a model has not previously been theoretically considered.

It is known that the loss of an electron by the luminescence center can result in luminescence quenching. However, within the framework of the proposed model, it is possible not only to qualitatively explain the sharp downward jumps in the luminescence intensity due to quenching, but also the sharp upward jumps. Assuming that negatively charged / electrically neutral luminescence centers are donors / acceptors of electrons, localized at the ends of unwound fibers, we show that luminescence quenching occurs when these centers approach each other subject to non-polarized wave. At the same time, when the distance between these centers grows, the quenching effect is suppressed, and the luminescence is enhanced. Within the framework of this model, it is possible to formulate requirements for an external electromagnetic wave. First, this wave must be randomly polarized, that is, the electric field vector of the wave must have random directions across the unwound polymer fibers. In addition, this wave must have a low frequency so that it can move polymer fibers, which are sufficiently inertial. Finally, this wave must not be absorbed by the water. The last two requirements seem to be satisfied by a wave at a frequency of 100 MHz. Therefore, carrying out experiments with radiation at other frequencies, apparently, does not make sense. The source of radiation is a whip antenna, that is, the wave from such a source is linearly polarized (the vector E is directed along the antenna, and this direction is maintained within the distance between the antenna and the experimental setup according to photoluminescence spectroscopy).

The question arises, how to achieve random polarization of the incident wave on the surface of a polymer membrane? This could be realized due to the scattering of linearly polarized radiation by anisotropic particles. In fact, we in our experiment show for the first time that submicron solid particles scatter radiation at a frequency of 100 MHz. We agree with the reviewer that the submicron particles we use are not well characterized. The most typical patterns of such particles, obtained with tunneling electron microscope, are presented. These photographs show that these particles play the role of isotropic (spherically symmetric) or anisotropic Rayleigh scatterers with respect to radiation at a frequency of 100 MHz; the size of these particles is much less than the wavelength. A more detailed characterization, in our opinion, is not necessary to explain the results obtained. Summing up, we found a number of new effects arisen due to unwinding of polymer fibers into the bulk of the surrounding liquid providing that the membrane is irradiated with a low-frequency non-polarized electromagnetic wave. We have completely rewritten the Conclusion section, which includes the above comments.

Reviewer 2 Report

This work reported the impact of low-frequency radiation on polymer membrane in water. When different nanostructure samples are added to water, photoluminescence spectrum have some interesting behavior. It seems that anisotropic plays a role in the phenomenon. A theoretical model is proposed to explain the experiment.

This paper is well written and organized, may be accepted after minor revision as follow,

1.     Are there other experimental methods can help enhance author’s explanation of photoluminescence phenomenon? Photoluminescence is a common method, but more evidence can make the explanation more solid.

2.     Can authors try other materials’ nanostructure besides selenium and gold?

3.     Would authors try other shape for selenium and gold, besides nanoparticles and cylinders? As anisotropic may play a role in the phenomenon, different levels of anisotropic of sample may help us get a better understanding of the structure.

4.     Can electric field or magnetic field affect the dynamics of luminescence?

Author Response

We are grateful to the reviewer for an attentive reading of our manuscript. The manuscript was rewritten in accordance with the reviewer' advices and comments. In what follows we present our replies to the particular reviewer' remarks. Our replies are highligthed with italic font.

  1. Are there other experimental methods can help enhance author’s explanation of photoluminescence phenomenon? Photoluminescence is a common method, but more evidence can make the explanation more solid.

In our opinion, no additional explanation of the photoluminescence effect is required, since this effect is well enough described in terms of quantum physics. The specificity of photoluminescence excitation in our case is that the luminescence centers (sulfonate groups) are localized at the ends of polymer fibers unwound into the bulk of the liquid. These sulfonate groups can be electrically charged or neutral. We show that if we consider luminescence centers as donors and acceptors of electrons, that is, if we take into account the possibility of electron transferring from a donor to an acceptor, then it becomes possible to control the luminescence dynamics when the membrane is irradiated with a non-polarized low-frequency electromagnetic wave.

  1. Can authors try other materials’ nanostructure besides selenium and gold?

We investigated gold and selenium in our experiments. As is known, gold belongs to the class of metals, while selenium belongs to the class of non-metals. A characteristic feature of non-metals is a larger (compared to metals) number of electrons at the external energy level. We were interested to find out whether any specific features would appear when a low-frequency electromagnetic wave is scattered by metal and non-metal submicron particles. However, no significant differences were found in scattering excitation. Therefore, we did not study other materials.

  1. Would authors try other shape for selenium and gold, besides nanoparticles and cylinders? As anisotropic may play a role in the phenomenon, different levels of anisotropic of sample may help us get a better understanding of the structure.

Thank you for this question. We used two different experimental protocols of laser ablation of solid targets for manufacturing submicron particles. In these experiments, either spherically symmetrical or elongated (non-symmetrical) particles are obtained. Unfortunately, in experiments on laser ablation, there is no technique for obtaining nanoparticles of a given shape.

  1. Can electric field or magnetic field affect the dynamics of luminescence?

This is a very good question. Unfortunately, we have not yet carried out experiments on the effect of electric and magnetic fields on the luminescence dynamics. Therefore, it is difficult for us to give some comments.